# Glyphosate Use in the European Agricultural Sector and a Framework for Its Further Monitoring

**Clémentine Antier [1], Per Kudsk [2], Xavier Reboud [3], Lena Ulber [4], Philippe V. Baret [1,*] and Antoine Messéan [5]**

[1]   Sytra, Earth and Life Institute, Université catholique de Louvain, 1348 Ottignies-Louvain-la-Neuve, Belgique; clementine.antier@uclouvain.be

[2]   Department of Agroecology, Aarhus University, DK-4200 Slagelse, Denmark; per.kudsk@agro.au.dk

[3]   Agroécologie, AgroSup Dijon, INRAE, University of Bourgogne Franche-Comté, F-21000 Dijon, France; xavier.reboud@inrae.fr

[4]   Institute for Plant Protection in Field Crops and Grassland, Julius Kühn-Institut (JKI), Messeweg 11-12, 38104 Braunschweig, Germany; lena.ulber@julius-kuehn.de

[5]   Eco-Innov, INRAE, 78850 Thiverval-Grignon, France; antoine.messean@inrae.fr

*   Correspondence: philippe.baret@uclouvain.be

**Abstract:** Monitoring pesticide use is essential for assessing farming practices and the risks associated with the use of pesticides. Currently, there are neither consolidated, public data available on glyphosate use in Europe, nor a standardized categorization of its major uses. In this study, data on glyphosate sales and use in Europe were collected from multiple sources and compiled into a dataset of the agricultural use of glyphosate from 2013 to 2017. The survey shows that glyphosate represented 33% of the herbicide volume sold in Europe in 2017. One third of the acreage of annual cropping systems and half of the acreage of perennial tree crops received glyphosate annually. Glyphosate is widely used for at least eight agronomic purposes, including weed control, crop desiccation, terminating cover crops, terminating temporary grassland and renewing permanent grassland. Glyphosate use can be classified into occasional uses—i.e., exceptional applications, triggered by meteorological conditions or specific farm constraints—and recurrent uses, which are widespread practices that are embedded in farming systems and for which other agronomic solutions may exist but are not frequently used. This article proposes a framework for the precise monitoring of glyphosate use, based on the identification of the cropping systems in which glyphosate is used, the agronomic purposes for which it is employed, the dose used and the rationale behind the different uses.

**Keywords:** agriculture; glyphosate; herbicide; pesticide use; pesticide dependency; EU

## 1. Introduction

Glyphosate is the most widely used herbicide in the world [1,2]. The global sales volume was estimated at 825,804 t in 2014—of which 746,580 t (90%) were used by the agricultural sector [3], the rest being used in non-crop areas. This volume accounted for 18% of the 4,105,783 t of pesticide active ingredients (a.i.) and 92% of the 814,614 t of herbicide a.i. sold to the agricultural sector globally in 2014, according to the Food and Agriculture Organization (FAO) and the Organisation for Economic Co-operation and Development (OECD) [4,5].

First marketed in 1974 [2], the sales of glyphosate have steadily increased: in 1994, the use of glyphosate by the agricultural sector was estimated at 56,296 t of active ingredient; in 2000, it reached 155,367 t; and in 2010, it was 578,124 t [3]. As shown by Benbrook [3], multiple factors have driven the increase in glyphosate use at the global level, including an increase in the area treated with the

herbicide, the marketing of glyphosate-tolerant crop varieties, the increasing number of authorized glyphosate uses in different crops, the adoption of no-till and conservation tillage systems which rely on herbicides (especially in the USA and South America), the declining market price of glyphosate [6] and new application methods. Glyphosate is now used extensively both in annual cropping systems and in perennial crops.

In parallel with the increasing use of glyphosate, controversies have arisen in Europe (as well as in the USA, Latin America and Asia) regarding the direct and indirect effects of glyphosate use on the environment and human health [7,8] and regarding glyphosate registration and regulations [9–13]. In Europe, debates have taken place in multiple arenas (e.g., social networks, scientific journals and public policy decision arenas) and have led, for example, to a European Citizens' Initiative (ban glyphosate and protect people and the environment from toxic pesticides) [14], as well as the decision by the Austrian parliament to ban glyphosate [15,16] and to a statement by the German government that glyphosate use should be significantly reduced by 2023 [17,18]. In addition, the wide use of glyphosate is linked to the development of resistant weeds, including in Europe [19,20].

Monitoring pesticide use is essential to providing an overview of current farming practices, assessing the level of risk associated with pesticide use, designing relevant agricultural policies and envisaging further agronomical research and extension. Although glyphosate use is widely discussed in terms of its toxicological and environmental impacts, little information is available on the quantities of glyphosate used in EU countries. Neither FAO, OECD or Eurostat public statistics provide glyphosate sales or use volumes at the national or regional level. The consultation of major scientific databases (Google Scholar, ScienceDirect) with the keywords "glyphosate", "herbicide use" and "pesticide use" lead to only a few articles reporting global glyphosate sales: one article by Woodburn provides data until 1997 [6] and one article by Benbrook provides data until 2014 [3]; no data on glyphosate global use in more recent years can be found. This lack of publicly accessible data on glyphosate use has already been underlined by the non-governmental organization Friends of the Earth [21] and by the Pesticide Action Network [22] at the European level as well as by Benbrook [3] at the global level. Against this background, the ENDURE network (www.endure-network.eu) launched a survey in 2019 to gather data on the uses of glyphosate and the existing alternatives to glyphosate use in European countries.

This paper provides a comprehensive database on glyphosate use in the EU and some non-EU countries for the entire agricultural sector as well as for different crop types [23]. A framework is proposed for understanding and better standardizing the future monitoring of glyphosate use. Factors that could explain the different level of use across countries are assessed, and potential drivers of the growing use of glyphosate in Europe are discussed.

## 2. Materials and Methods

### 2.1. Data Collection Process and Scope

Data was collected in a two-step procedure relying on the ENDURE network, a network of researchers and public policy institutions as initial points of contact. First, each point of contact (1–3 points of contact per country) received a questionnaire and was asked to provide national data regarding the volume and uses of glyphosate in their country, obtained either from public statistics or values estimated by national experts. Next, the results were analyzed and returned to the points of contact for a further review and data check.

The survey covered all EU28 countries plus Norway, Serbia, Switzerland and Turkey. Datasets were generated either for EU28, EU28+3 (EU28 + Norway, Switzerland and Turkey) or EU28+4 (EU28 + Norway, Serbia, Switzerland and Turkey). The following information was collected: total annual glyphosate sales in each country from 2013 to 2017, the share of glyphosate sales and total pesticide sales linked to the agricultural sector, the percentage of the acreage of individual crops treated annually with glyphosate (annual crops: maize, oilseed rape and winter wheat fields; perennial crops: vineyards, fruit orchards and olive orchards; and permanent and temporary grasslands), the average volumes

of glyphosate used per hectare and the agronomic purposes for which glyphosate is used in each crop. Throughout the paper, all the data on volumes of applied glyphosate refer to kilograms (when reporting per ha) or tonnes (when reporting total volumes at the national or European level) of the glyphosate active ingredient (encompassing all the commercial end-user formulations). Globally, there are more than 750 herbicide products containing glyphosate available for sale [24].

## 2.2. Glyphosate National Sales Data

Glyphosate national sales data in the period 2013–2017 were collected in two steps. First, the glyphosate sales data were gathered from 25 countries (Table 1), covering 88% of the conventional farming area in the EU28+4. This made it possible to calculate the ratio of glyphosate to total herbicide sales in each country. Second, for the seven countries for which no sales data could be obtained, an estimate was calculated. This estimate was based on the total volume of herbicide a.i. sold in each country (provided by Eurostat) and the average ratio of glyphosate to total herbicide sales in EU countries for which data was available and collected during step one. The estimated figures were reviewed and confirmed by national points of contact to ensure consistency with local knowledge. In total, the data from the ENDURE survey account for more than 90% of the estimated EU28+4 glyphosate sales volume, while complementary estimates account for less than 10%.

**Table 1.** Overview of the data acquired through the survey for the EU28+4 countries.

| | Share of Glyphosate Volume Sold to the Agricultural Sector [1] | Data Source [2] | Share of Pesticide Volume Sold to the Agricultural Sector [1] | Data Source [2] | Glyphosate Sales from 2013 to 2017 [1] | Data Source [2] |
|---|---|---|---|---|---|---|
| Austria | x | S1 | | | xx | S1 |
| Belgium | x | S2 | x | S2 | xx | S1 |
| Bulgaria | | | | | | E |
| Croatia | x | S2 | | | xx | S1 |
| Cyprus | | | | | | E |
| Czech Rep. | | | | | xx | S1 |
| Denmark | x | S1 | x | S2 | xx | S1 |
| Estonia | | | x | S2 | x | S1 |
| Finland | x | S2 | x | S2 | xx | S1 |
| France | x | S1 | | | xx | S1 |
| Germany | | | | | xx | S1 |
| Greece | | | | | x | S2 |
| Hungary | x | S2 | x | S2 | xx | S1 |
| Ireland | | | | | | E |
| Italy | | | | | xx | S1 |
| Latvia | | | x | S2 | xx | S1 |
| Lithuania | x | S2 | | | xx | S1 |
| Luxembourg | | | | | | E |
| Malta | | | | | | E |
| Netherlands | x | S2 | | | x | S2 |
| Norway | x | S2 | x | S2 | xx | S1 |
| Poland | | | | | xx | S1 |
| Portugal | | | | | xx | S1 |
| Romania | | | | | | E |
| Serbia | | | | | x | S2 |
| Slovakia | | | | | | E |
| Slovenia | | | | | xx | S1 |
| Spain | | | x | S2 | xx | S2 |
| Sweden | x | S1 | x | S1 | xx | S1 |
| Switzerland | | | | | xx | S1 |
| Turkey | x | S2 | | | xx | S1 |
| UK | x | S1 | x | S2 | x | S1 |

[1] xx indicates that the data was provided for each year from 2013 to 2017; x indicates that data was provided for some years; empty cell indicates that no data was provided (an estimate was therefore calculated afterwards).
[2] S1 indicates data obtained through the ENDURE survey based on national statistics; S2 indicates data obtained through the ENDURE survey based on an assessment by national experts when data was not available in national statistics; E indicates glyphosate use in the context of annual cropping systems and perennial crops.

### 2.3. Total Pesticide and Herbicide Sales Data

In addition to the data from the survey, the national pesticide and herbicide sales and the respective areas used for agricultural production were obtained from the Eurostat database. At the EU level, active pesticide ingredients are classified according to their targets (herbicide, insecticide, fungicide, etc.) and chemical composition [25]. Sales volumes are recorded annually in Eurostat, with the 2017 data being the most recent data available at the time of the analysis. Although Eurostat provides total pesticide sales for each country, data are not reported for the individual market sectors (agriculture, forestry, railways and infrastructure, industrial uses, non-professional uses, etc.).

### 2.4. Share of Glyphosate and Total Pesticides Used in the Agricultural Sector

Data on the share of pesticide use in the agricultural sector versus its use in other sectors were collected during the survey in case this information was not provided in the Eurostat public database. This information could be obtained for ten countries (Belgium, Denmark, Estonia, Finland, Hungary, Latvia, Norway, Spain, Sweden and the UK). Similarly, the data on the share of glyphosate sales for the agricultural sector was collected during the survey. The information could be obtained for 13 countries (Austria, Belgium, Croatia, Denmark, France, Finland, Hungary, Lithuania, Netherlands, Norway, Sweden, the UK and Turkey) (Table 1). The data provided were based either on national statistics or expert assessments.

### 2.5. Average Use of Glyphosate Per Hectare (AUG)

The average use of glyphosate (AUG) in the agricultural sector at the national and European level was compared to the acreage under conventional agriculture where glyphosate can potentially be applied. The conventionally managed agricultural acreage was calculated as the total national utilized agricultural area (UAA), excluding the acreage under organic certification (Eurostat data). This national-level indicator is complementary to field level indicators that may be used for the characterization of farmers' practices. The AUG indicator is particularly relevant for comparisons between countries and useful for monitoring the long-term evolution of glyphosate use in Europe.

### 2.6. Glyphosate Use in the Context of Annual Cropping Systems and Perennial Crops

Data on the acreage of each crop annually receiving glyphosate were collected during the survey. Three of the major European annual crops (wheat, maize and rape) and perennial tree crops (olive groves, vineyards and fruit orchards) as well as temporary grasslands were included in the survey in order to cover a diversity of situations and a significant proportion of the European acreage. Together, these six crops and temporary grassland covered 36% of the total EU28 UAA in 2017 (Eurostat). The proportion of each crop's acreage treated with glyphosate at the national and European levels was then calculated. In total, the survey covered an acreage of 49–99% of the respective crop or grassland acreage in the EU28+4 countries (Table 2).

**Table 2.** Acreage of each of the crops included in the study in the EU28+4 in 2017 and the share of the acreage covered by the survey.

| Crop | Acreage of the Crop at the EU28+4 Level [1] (1000 ha) | Countries in Which Data Was Available | Acreage Covered by the Survey (1000 ha) | Share of the EU28+4 Crop Acreage Covered by the Survey (%) |
|---|---|---|---|---|
| **Annual crops** | | | | |
| Wheat [2] | 34,246 | Austria, Belgium, Croatia, Estonia, Finland, France, Germany, Hungary, Ireland, Latvia, Lithuania, Norway, Portugal, Serbia, Spain, Sweden, Switzerland, UK | 16,819 | 49% |

**Table 2.** *Cont.*

| Crop | Acreage of the Crop at the EU28+4 Level [1] (1000 ha) | Countries in Which Data Was Available | Acreage Covered by the Survey (1000 ha) | Share of the EU28+4 Crop Acreage Covered by the Survey (%) |
|---|---|---|---|---|
| **Annual crops** | | | | |
| Maize [3] | 16,679 | Austria, Belgium, Croatia, Estonia, France, Germany, Hungary, Ireland, Latvia, Lithuania, Portugal, Serbia, Switzerland, Turkey, UK | 9883 | 59% |
| Rape [4] | 6808 | Austria, Belgium, Estonia, Finland, France, Germany, Hungary, Ireland, Latvia, Lithuania, Norway, Switzerland, UK | 4088 | 60% |
| **Perennial crops** | | | | |
| Olive groves [5] | 5897 | Croatia, Greece, Italy, Portugal, Spain, Turkey | 5867 | >99% |
| Vineyards [6] | 3118 | Austria, Belgium, Croatia, France, Germany, Greece, Hungary, Italy, Portugal, Serbia, Spain, Switzerland, Turkey, UK | 2871 | 92% |
| Fruit orchards [7] | 5436 | Austria, Belgium, Croatia, Finland, France, Germany, Greece, Hungary, Ireland, Latvia, Lithuania, Norway, Portugal, Serbia, Spain, Sweden, Switzerland, Turkey, UK | 4316 | 79% |
| **Grasslands** | | | | |
| Temporary grassland | 9180 [1] | Austria, Finland, France, Ireland, Sweden, Switzerland, UK | 6065 | 66% |

[1] The data covers the EU28+4 except for temporary grassland, which covers the EU28+2 (EU28 + Serbia, Switzerland), as no data were available in Eurostat for Turkey and Norway; [2] wheat and spelt (code C 1100 in Eurostat); [3] grain maize and corn-cob-mix (C1500) and green maize (G3000); [4] rape and turnip rape seeds (I1110); [5] olives (O1000 in Eurostat); [6] grapes for wines (W1100); [7] fruits, berries and nuts (excluding citrus fruits, grapes and strawberries) (F000) and citrus fruits (T000).

## 3. Results

### 3.1. Glyphosate Sales and Average Use in Europe

Based on the survey data and complementary estimates, the total volume of glyphosate sold at the EU28+4 level was 49,427 t in 2017. Between 2013 and 2017, the total volumes of glyphosate sold in the EU28+4 fluctuated between 46,981 and 49,427 t with no clear trend. In 2017, the glyphosate sales were highest in France (19% of the total EU28+4 glyphosate sales by volume), Poland (13%), Germany (9%), Italy (7%), Spain (7%) and Serbia (6%) (Table 3, Figure A1 in Appendix B).

Overall, glyphosate represented 33% of the total herbicide a.i. sales at the EU28+3 level in 2017. However, glyphosate was not the most widely used herbicide in all the EU countries. Indeed, the proportion of glyphosate sales compared to the national total herbicide sales in 2017 varied from 15% to 69% in the countries surveyed. Glyphosate accounted for more than half of the herbicide active ingredients sold in six countries (Estonia, Finland, Greece, Italy, Norway and Portugal), for 20% to 50% in 17 countries (Austria, Belgium, Croatia, Czech Republic, Denmark, France, Germany, Hungary, Latvia, Lithuania, Netherlands, Poland, Slovenia, Spain, Sweden, Switzerland and UK) and less than 20% in one country (Turkey) (Figure A2 in Appendix B).

**Table 3.** Total glyphosate sales (tonnes of a.i.) in the EU28+4 countries from 2013 to 2017 based on the ENDURE survey and complementary estimates.

| | 2013 | 2014 | 2015 | 2016 | 2017 | Shift from 2013 to 2017 (%) | Share of Total EU28+4 Sales in 2017 | Data Source [1] | Herbicide Sales in 2017 (t of a.i.) | Proportion of Glyphosate Compared to All Herbicides in 2017 (%) |
|---|---|---|---|---|---|---|---|---|---|---|
| Austria | 174 | 338 | 327 | 312 | 329 | 89% | 1% | S | 1297 | 25% |
| Belgium | 587 | 596 | 512 | 503 | 619 | 5% | 1% | S | 2334 | 27% |
| Bulgaria | 261 | 242 | 236 | 744 | 629 | 141% | 1% | E | 1699 | 38% |
| Croatia | 231 | 302 | 285 | 268 | 217 | −6% | <1% | S | 669 | 32% |
| Cyprus | 57 | 57 | 57 | 58 | 51 | −9% | <1% | E | 139 | 38% |
| Czech Rep. | 935 | 859 | 698 | 772 | 751 | −20% | 2% | S | 2562 | 29% |
| Denmark [2] | 1371 | 610 | 842 | 1126 | 1241 | −9% | 3% | S | 2485 | 50% |
| Estonia | nd | 277 | nd | 412 | 253 | nd | 1% | S | 463 | 55% |
| Finland | 550 | 710 | 860 | 840 | 660 | 20% | 1% | S | 1006 | 66% |
| France | 9370 | 10,070 | 9110 | 9110 | 9324 | 0% | 19% | S | 30,230 | 31% |
| Germany | 5065 | 5426 | 4797 | 3780 | 4694 | −7% | 9% | S | 16,706 | 28% |
| Greece | nd | nd | nd | nd | 1300 | nd | 3% | S | 1674 | 78% |
| Hungary | 885 | 1296 | 1423 | 1769 | 1647 | 86% | 3% | S | 4270 | 39% |
| Ireland | 742 | 755 | 777 | 831 | 674 | −9% | 1% | E | 1820 | 38% |
| Italy | 4566 | 4504 | 4460 | 4225 | 3699 | −19% | 7% | S | 7114 | 52% |
| Latvia | 153 | 178 | 181 | 207 | 168 | 10% | <1% | S | 801 | 21% |
| Lithuania | 502 | 470 | 502 | 422 | 253 | −50% | 1% | S | 1252 | 20% |
| Luxembourg | 31 | 33 | 31 | 23 | nd | nd | <1% | E | nd | 38% |
| Malta | 3 | 3 | 2 | 2 | 1 | −66% | <1% | E | 2 | 38% |
| Netherlands [3] | nd | nd | nd | nd | 742 | nd | 2% | S | 2902 | 37% |
| Norway | 299 | 300 | 355 | 346 | 299 | 0% | 1% | S | 467 [2] | 64% |
| Poland | 5056 | 4992 | 4397 | 5392 | 6665 | 32% | 13% | S | 13,655 | 49% |
| Portugal | 1120 | 1687 | 1459 | 1307 | nd | nd | 3% | S | 1899 | 69% |
| Romania | 2235 | 1861 | 2353 | 1877 | 2032 | −9% | 4% | E | 5486 | 38% |
| Serbia | nd | nd | nd | nd | 2900 | nd | 6% | S | nd | nd |
| Slovakia | 429 | 450 | 451 | 400 | 409 | −5% | 1% | E | 1105 | 38% |
| Slovenia | 51 | 73 | 73 | 92 | 86 | 68% | <1% | S | 235 | 36% |
| Spain | 2879 | 2883 | 3120 | 3787 | 3633 | 26% | 7% | S | 16,077 | 23% |
| Sweden | 632 | 626 | 683 | 657 | 485 | −23% | 1% | S | 1731 | 28% |
| Switzerland | 308 | 296 | 228 | 204 | 189 | −39% | <1% | S | 599 | 32% |
| Turkey | 1659 | 1698 | 1709 | 1755 | 1789 | 8% | 4% | S | 11,825 | 15% |
| UK | 1494 | 1911 | 1927 | 2240 | nd | nd | 5% | S | 9682 | 23% |
| **EU28** [4] | **41,814** | **43,355** | **42,000** | **43,319** | **44,250** | **+6%** | **90%** | | **129,359** | **34%** |
| **EU28+3** [4] | **44,081** | **45,649** | **44,292** | **45,624** | **46,527** | **+6%** | **94%** | | **142,251** | **33%** |
| **EU28+4** [4] | **46,981** | **48,549** | **47,192** | **48,524** | **49,427** | **+5%** | **100%** | | **nd** | **nd** |

[1] S: ENDURE survey, 2019; E: complementary estimates; when no data was available through the survey (numbers in grey: Bulgaria, Cyprus, Ireland, Luxembourg, Malta, Romania and Slovakia), an estimate is offered, based on the total volume of herbicides sold in each country and the average share that glyphosate represents among herbicides in European countries for which data was available (an average of 38% of the volume of herbicides sales). [2] The decrease in 2014 and 2015 was due to a change in the taxation of pesticides, rendering glyphosate more expensive. [3] Sales to the agricultural sector and private entities; sales to other sectors (railways, etc.) are not included. [4] When no 2017 data were available, data from the closest year was used. nd stands for no data.

*3.2. Glyphosate Sales to the Agricultural Sector*

According to the survey, an average of 90% of the total pesticide sales (by volume) were used by the agricultural sector. Similarly, an average of 90% of the glyphosate sales (by volume) were sold to the agricultural sector. This is consistent with Benbrook [3], who indicated that the share of the agricultural sector in glyphosate global sales was 89% in 2010, 90% in 2012 and 90% in 2014.

Across countries, the average use of glyphosate per hectare (AUG) varied from 0.04 to 0.56 kg per ha (Table 4). The AUG in the EU28+3 countries was 0.20 kg of glyphosate in 2017, while the total use of herbicides per hectare was estimated at 0.62 kg of a.i (Table A1 in Appendix A).

**Table 4.** Glyphosate use in the agricultural sector (annual sales and average per hectare of utilized agricultural area (UAA)) in the EU28+4 countries in 2017.

| | Glyphosate Sales in 2017 [2] (Tonnes of a.i.) | Data Source of Glyphosate Sales [2] | Estimated Share of Glyphosate Sales Used by the Agriculture Sector [3] | UAA in Conventional Agriculture [1] (1000 ha) | Average Use of Glyphosate in the Agriculture Sector (AUG) (kg of a.i./ha) |
|---|---|---|---|---|---|
| Austria | 329 | S | 98% | 2035 | 0.16 |
| Belgium | 619 | S | 60% | 1246 | 0.30 |
| Bulgaria | 649 | E | *90%* | 4893 | 0.12 |
| Croatia | 217 | S | 85% | 1400 | 0.13 |
| Cyprus | 53 | E | *90%* | 116 | 0.40 |
| Czech Rep. | 751 | S | *90%* | 3025 | 0.22 |
| Denmark | 1241 | S | 99% | 2405 | 0.51 |
| Estonia | 253 | S | *90%* | 806 | 0.28 |
| Finland | 660 | S | 95% | 2014 | 0.31 |
| France | 9324 | S | 95% | 27,357 | 0.32 |
| Germany | 4694 | S | *90%* | 15,549 | 0.27 |
| Greece | 1300 | S | *90%* | 4742 | 0.25 |
| Hungary | 1647 | S | 97% | 5153 | 0.31 |
| Ireland | 695 | E | *90%* | 4415 | 0.14 |
| Italy | 3699 | S | *90%* | 10,935 | 0.30 |
| Latvia | 168 | S | *90%* | 1663 | 0.09 |
| Lithuania | 253 | S | 90% | 2701 | 0.08 |
| Luxembourg | 23 [4] | E | *90%* | 126 | 0.16 |
| Malta | 1 | E | *90%* | 12 | 0.07 |
| Netherlands | 742 | S | *90%* | 1734 | 0.39 |
| Norway | 299 | S | 95% | 935 | 0.30 |
| Poland | 6665 | S | *90%* | 14,003 | 0.43 |
| Portugal | 1307 [4] | S | *90%* | 3349 | 0.35 |
| Romania | 2095 | E | *90%* | 13,119 | 0.14 |
| Serbia | 924 | S | *90%* | 3425 | 0.24 |
| Slovakia | 422 | E | *90%* | 1722 | 0.21 |
| Slovenia | 86 | S | *90%* | 435 | 0.18 |
| Spain | 3633 | S | *90%* | 21,759 | 0.15 |
| Sweden | 485 | S | 97% | 2435 | 0.19 |
| Switzerland | 189 | S | *90%* | 1368 | 0.12 |
| Turkey | 1789 | S | 85% | 37,760 | 0.04 |
| UK | 2240 [4] | S | 80% | 16,974 | 0.11 |
| **Total EU28** | **44,250** | **S, E** | **91%** | **166,120** | **0.24** |
| **Total EU28+3** | **46,527** | **S, E** | **91%** | **206,184** | **0.20** |
| **Total EU28+4** | **49,427** | **S, E** | **91%** | **209,608** | **0.21** |

[1] Eurostat: where no data were available for 2017, the gaps were filled using data from the closest year, with preference given to the previous year when available; the herbicide sales volume in Luxembourg is from 2016 and in Norway is from 2015. [2] ENDURE survey (S, in black) or estimate (E, in grey). Total sales, all sectors included. [3] Countries for which data were obtained though the ENDURE survey: Austria, Belgium, Croatia, Denmark, Finland, France, Hungary, Lithuania, Netherlands, Norway, Sweden, Turkey and the UK. For other countries (in italics), an average ratio of 90% is used. [4] Data from 2016 instead of 2017.

### 3.3. Use of Glyphosate in Annual Cropping Systems

The indicator used in our study was the percentage of each crop acreage treated with glyphosate at least once a year. Across the countries for which data was available, 25–52% of the annual crop acreage (including wheat, rape and maize) was treated with glyphosate annually (Table 5). The use of glyphosate in annual crops (which includes terminating previous cover crops, controlling weeds pre-sowing or on stubble and as a harvest aid; see below in Figure 1) varied greatly among the countries (Table 6). Based on the survey, the glyphosate use in maize fields was comparatively low in several EU countries (<10% of the maize acreage in Belgium, Croatia, Estonia, Hungary, Lithuania and Portugal), while glyphosate was used on more than 30% of the maize area in other countries (France, UK and Serbia). In oilseed rape, the percentage of the crop area receiving glyphosate varied from less than 10% (Austria and Belgium) to more than 60% (Germany, Ireland and UK). Similarly, the percentage of the winter wheat area receiving glyphosate varied from less than 10% (Austria and Portugal) to more than 45% (Finland, Lithuania, Serbia and UK) and even up to 90% (Hungary). The average glyphosate use rate ranged from 0.40 to 2.70 kg a.i./ha. (Table 6).

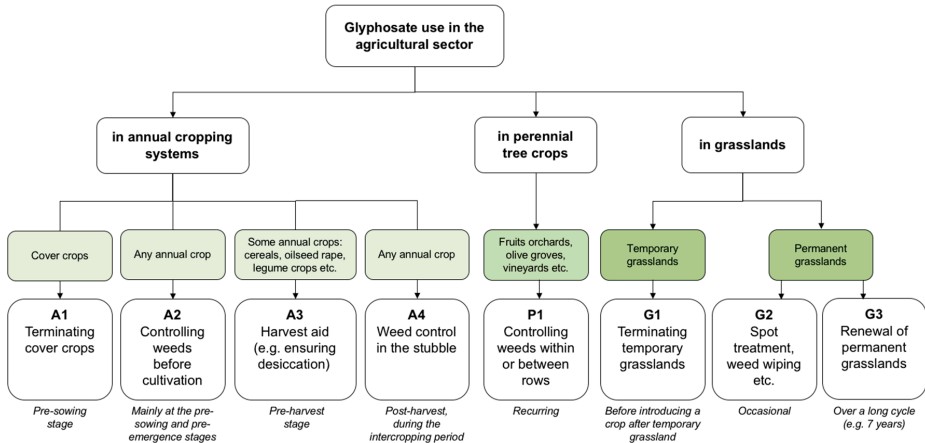

**Figure 1.** A framework for analyzing glyphosate uses: types of crop for which glyphosate is used and major agronomic purposes. Source: ENDURE survey. Notes: other types of crops in which glyphosate is used include tree plantations and seed production fields [26]. In other geographical areas with GMO crops, such as the USA, glyphosate is also used within glyphosate-tolerant crops; this use could be identified as "A5". A2 includes all uses of glyphosate for controlling weeds before cultivation, including in the context of false seeding techniques. A further use of glyphosate is also its application with selective equipment (avoiding crop contact) at the post-emergence stage to control weeds between crop rows.

**Table 5.** Acreage of each crop at the EU28+4 level in 2017, acreage treated with glyphosate and average use per hectare within the scope of the survey.

| Crop | Total Acreage of the Crop at EU28+4 Level [1] (1000 ha) | Acreage Covered by the Survey (1000 ha) | Acreage Treated with Glyphosate within the Scope of the Survey (1000 ha) | Area Not Treated with Glyphosate within the Scope of the Survey (1000 ha) | Range of Glyphosate Dose per Hectare (kg of a.i./ha) |
|---|---|---|---|---|---|
| **Annual cropping systems** | | | | | |
| Wheat [2] | 34,246 | 16,819 | 5425 32% | 11,394 68% | 0.50–2.00 |
| Maize [3] | 16,679 | 9883 | 2467 25% | 7417 75% | 0.60–2.70 |
| Rape [4] | 6808 | 4088 | 2127 52% | 1961 48% | 0.40–2.00 |
| **Perennial crops** | | | | | |
| Olive groves [5] | 5897 | 5867 | 2631 45% | 3236 55% | 1.67–2.50 |
| Vineyards [6] | 3118 | 2871 | 906 29% | 1965 63% | 0.20–2.50 |
| Fruit orchards [7] | 5436 | 4316 | 2778 64% | 1538 36% | 0.45–2.50 |
| **Grasslands** | | | | | |
| Temporary grassland | 9180 [1] | 6065 | 1171 19% | 4894 81% | no data |

[1] Except when mentioned specifically, for temporary grassland, the scope is EU28+2 (EU28 + Serbia, Switzerland); no data available in Eurostat for Turkey and Norway. [2] Wheat and spelt (code C 1100 in Eurostat); [3] grain maize and corn-cob-mix (C1500) and green maize (G3000); [4] rape and turnip rape seeds (I1110); [5] olives (O1000 in Eurostat); [6] grapes for wines (W1100); [7] fruits, berries and nuts (excluding citrus fruits, grapes and strawberries) (F000) and citrus fruits (T000).

Reading note example: regarding temporary grassland, the total acreage at the EU28+2 level is 9.18 million ha. The survey provided data in seven countries (Table 2), covering 6065 thousand ha—i.e., 66% of the EU28+2 temporary grassland acreage. Within the scope of the survey, 1171 thousand ha (19%) of temporary grassland was treated with glyphosate and 4894 thousand ha (81%) was not treated with glyphosate.

**Table 6.** Share of annual crop acreage treated with glyphosate and the average use in kg a.i. per hectare in 20 of the EU28+4 countries.

| | Maize | | Oilseed Rape [1] | | Winter Wheat [2] | |
|---|---|---|---|---|---|---|
| Purposes for Which Glyphosate May Be Used | Includes Terminating Previous Cover Crops (A1); Controlling Weeds Pre-Sowing or on Stubble (A2, A4); Ensuring Desiccation or More Generally Using Glyphosate as a Harvest Aid [3] (A3) | | | | | |
| | Percentage of the Crop Area Treated with Glyphosate (I6) | Average Dose Per Year [4] kg of a.i./ha (I7b) | Percentage of the Crop Area Treated with Glyphosate (I6) | Average Dose Per Year [4] kg of a.i./ha (I7b) | Percentage of the Crop Area Treated with Glyphosate (I6) | Average Dose Per Year [4] kg of a.i./ha (I7b) |
| Austria | 15% | 1.50 | 3% | 1.50 | 5% | 1.50 |
| Belgium [5] | 8% | 0.72–1.08 | 5% | 0.72–1.08 | 17% | 0.72–1.08 |
| Croatia | 5% | 1.80 | nd | nd | >20% | 1.80 |
| Estonia | 7% | 2.70 | 17% | 1.60 | 25% | 1.80 |
| Finland | na | na | 10% | 1.08 | 50% | 0.70–1.08 |
| France [6] | 35% | 0.60 | 35% | 0.40 | 20% | 0.50 |
| Germany | 28% | nd | 72% | nd | 31% | nd |
| Hungary | 1% | 1.80 | 40–50% | 1.40 | >90% | 2.00 |
| Ireland | 22% | nd | 66% | nd | 30% | nd |
| Latvia [7] | 10% | nd | 22% | 1.20 | 25% | 1.15 |
| Lithuania | <5% | 1.08–2.00 | 50–60% | 1.08–2.00 | 50–60% | 0.72–1.08 |
| Netherlands | 5% | nd | nd | nd | 16% | nd |
| Norway | nd | nd | 20–40% | 1.08–1.44 | 20–40% | 1.08–1.44 |
| Portugal | 1% | nd | nd | nd | 2% | nd |
| Serbia | 40% | 1.00 | nd | nd | 40–50% | 1.80 |
| Spain | nd | nd | nd | nd | 27% | nd |
| Sweden | nd | nd | nd | nd | >11% | 0.99–1.22 |
| Switzerland | 22% | 1.60 | 25% | 1.70 | 14% | 1.70 |
| Turkey | 15% | 1.00 | nd | nd | nd | nd |
| UK | 36% | 0.81 | 71% | 1.01 | 48% | 0.71 |
| All countries [8] | 25% | | 52% | | 32% | |

[1] Oilseed rape: includes both winter and spring oilseed rape (depending on the country); [2] winter wheat: includes both soft wheat and durum wheat (depending on the country); [3] according to the survey, the use of glyphosate for desiccation/harvest aid is authorised in Bulgaria, Croatia, Czech Republic, Denmark, Germany, Hungary, Ireland, Netherlands, Poland, Romania, Serbia, Spain, Sweden, Turkey and UK; [4] average in fields in which glyphosate is used; [5] data from the Flanders region; [6] data from sample; [7] proxy: percentage of surveyed fields; [8] in the scope of countries for which data was available (see Table 5). Data for maize, rape and wheat were not available in the remaining countries of the EU28+4. This table focuses on glyphosate use. Other herbicides may be used on the crop area and are not highlighted here. In Austria, for example, only 5% of the winter wheat crop area is treated with glyphosate, while 85% of the crop area is treated with another herbicide (ENDURE survey). In annual cropping systems, the allocation of glyphosate treatments to one crop or another may vary across countries (see Section 4.2). Figures should be refined before use for other studies or informing policy development. Year: Austria: 2017; Belgium: 2015; Croatia: 2017; Estonia: 2015; Finland: 2018; France: nd; Germany: 2013; Hungary: 2017; Ireland: 2012; Latvia: 2013–2017; Lithuania: 2017; Norway: 2014–2017; Portugal: 2017; Serbia: 2016; Spain: 2013; Switzerland: 2009–2017; Sweden: 2017; Turkey: nd; UK: 2017–2018. Note: nd: no data; na: not applicable. *A1* to *A4* refer to the numbering of glyphosate uses presented in Figure 1. *I6* and *I7b* refer to the numbering of indicators for monitoring glyphosate use presented below in Section 4.4.

## 3.4. Use of Glyphosate in Perennial Crops and Grasslands

Across the countries for which data was available, 39% of the fruit orchard acreage, 32% of the vineyard acreage and 45% of the olive grove acreage were treated with glyphosate annually. The area treated with glyphosate varied across countries from 20% to 92% of the national fruit orchard acreage, 13% to 95% of the national vineyard acreage and from 13% to 80% of the national olive grove acreage. The average glyphosate dose varied from 0.45 to 2.50 kg a.i./ha in fruit orchards, from 0.20 to 2.50 kg a.i./ha in vineyards and from 1.67 to 2.50 kg a.i./ha in olive groves (Table 7). In vineyards and fruit orchards treated with glyphosate, at least 50% and 65%, respectively, of the acreage treated received intra-row treatments only.

Across the countries, 19% of the temporary grassland acreage was treated with glyphosate annually (Table 7). In addition, the survey shows that glyphosate was also used on permanent grassland, for grassland renovation (undertaken once every seven years, for example) and for the localized or spot treatments of perennial weeds.

**Table 7.** Share of the perennial tree crops and temporary grassland acreage treated with glyphosate and average dose in the EU28+4 countries.

| | Vineyards | | Fruit Orchards | | Olive Groves | | Temporary Grassland |
|---|---|---|---|---|---|---|---|
| **Agronomic Purposes for Which Glyphosate May Be Used** | **Controlling Weeds between or within Rows (P1)** | | | | | | **Terminating Temporary Grassland (G1)** |
| | Percentage of the Crop Area Treated with Glyphosate (I8) | Average Dose Per Year [1] kg of a.i./ha (I10) | Percentage of the Crop Area Treated with Glyphosate (I8) | Average Dose Per Year [1] kg of a.i./ha (I10) | Percentage of the Crop Area Treated with Glyphosate (I8) | Average Dose Per Year [1] kg of a.i./ha (I10) | Percentage of the Crop Area Treated with Glyphosate (I8) |
| Austria | 55% | 1.80 | 40% | 1.80 | na | na | 1% |
| Belgium [2] | 50% | 0.72–1.08 | 92% | 0.72–1.08 | na | na | nd |
| Croatia | 80% | 2.50 | 70% | 2.50 | 80% | 2.50 | nd |
| Finland | na | na | 20% | 1.80 | na | na | 20% |
| France [3] | 36% | 0.20–1.00 | 90% | 0.81 | nd | nd | 30% |
| Germany | 60–80% | nd | 90% | nd | na | na | nd |
| Greece | 60% | nd | 70% | nd | 60% | nd | nd |
| Hungary | 90% | 2.00 | 92% | 2.00 | na | na | nd |
| Ireland | na | na | 49% | nd | na | na | 82% |
| Italy | 26% | nd | nd | nd | 13% | nd | nd |
| Latvia | nd | nd | 73% | 1.70 | nd | nd | nd |
| Lithuania | nd | nd | 90% | 1.30 | na | na | nd |
| Netherlands | nd | nd | 62% | nd | nd | nd | nd |
| Norway | na | na | 91% | 1.08–2.16 | na | na | nd |
| Portugal | 35% | nd | 21% | nd | 19% | nd | nd |
| Serbia | 50% | 1.80 | 50% | 1.80 | na | na | nd |
| Spain | 13% | nd | 33% | nd | 47% | nd | nd |
| Sweden | na | na | 61% | 0.57 | na | na | 4% |
| Switzerland | 83% | 0.90 | 60% | 0.70 | na | na | 24% |
| Turkey | 95% | 1.80 | 85% | 1.80 | 75% | 1.67 | nd |
| UK | 65% | 0.36 | 79% | 0.45 | na | na | <1% |
| All countries [4] | 32% | | 39% | | 45% | | |

[1] Average in fields in which glyphosate is used; [2] data from the Flanders region. [3] Data from fields sample; the average glyphosate dose is 0.70 kg a.i./ha/year and it varies significantly across systems (0.20–1.08 kg a.i./ha/year). [4] In the scope of countries for which data was available (see Table 5). This table focuses on glyphosate use. Other herbicide may be used on the crop area and are not highlighted here. In vineyards and fruit orchards treated with glyphosate, at least 50% and 65%, respectively, of the acreage treated received intra-row treatments only. This can partially explain the range of glyphosate doses reported. Year: Austria (vineyards): 2013; Austria (fruit orchards): 2017; Belgium: 2015; Croatia: 2017; Finland: 2018; France: nd; Germany: 2018; Hungary: 2017; Ireland: nd; Lithuania: 2017; Serbia: 2016; Spain: 2013; Sweden: 2017; Switzerland: 2009–2017; Turkey: nd; UK: 2016. nd: no data. na: not applicable. *P1* and *G1* refer to the numbering of glyphosate uses presented in Figure 1. *I8* and *I10* refer to the numbering of indicators for monitoring glyphosate use presented in Section 4.4.

## 4. Discussion

### 4.1. Comparison of the Use of Glyphosate in Europe and Globally

Until now, information on the volume of glyphosate sold to the agricultural sector at the EU level has not been available in the public domain. In the present study, data on the glyphosate use in Europe were estimated through a two-step survey, compiling national statistics, expert knowledge and estimates calculated using total herbicide sales. Given the absence of publicly available statistical data, this dataset is a good proxy for framing relevant research and policy orientations.

The volume of glyphosate sold at the EU28+4 level in 2014 to the agricultural sector was estimated at 48,549 t, which is 7% of the glyphosate volume sold globally to the agricultural sector (746,580 t [3]). Europe, therefore, is a minor market for this herbicide compared to other continents. In the same year, the USA accounted for 15% of the global glyphosate sales [3]. In comparison, in 1995 the global distribution of glyphosate use was 26% in North America, 22% in Latin America, 21% in the Far East, 18% in West Europe and 12% in the rest of the world [6]. This shows that glyphosate use has increased faster in other areas than in Europe and the USA.

At the EU28 level, the estimated glyphosate sales accounted for 34% of the total herbicide sales in 2017 (43,355 t of glyphosate a.i. vs. 129,359 t of the total herbicide a.i.). This is consistent

with the fact that 58% of the EU28 herbicide sales were classified under the chemical category H99 ("other herbicides"), which includes glyphosate (Appendix A). At the global level, glyphosate sales constitute a much higher proportion of total herbicide sales at around 92%. This significant difference in the proportion of glyphosate versus other herbicides in Europe and globally may be related to a) the widespread cultivation of glyphosate-tolerant crop varieties, especially in the USA and South America [27], which has resulted in a shift towards glyphosate use instead of other herbicides [3,28]; and b) the higher percentage of the agricultural acreage cultivated with no-till methods [29,30] that rely on glyphosate applications for weed control and seedbed preparation. In Europe, no-till methods are less common (5% of the European cropland area was under conservation agriculture in 2015/16, while 28% and 63% of the North America and South America cropland area, respectively, were under conservation agriculture [30]). In addition, innovative farmers groups are seeking to develop non-glyphosate-based conservation agriculture systems [31,32].

### 4.2. Reliability of Data Related to Annual Cropping Systems and Perennial Tree Crops

The survey provided an overview of the acreage of each crop treated with glyphosate in any given year, as well as the average dose rate per hectare in the countries surveyed. The use of glyphosate varied greatly among countries. The collection of national data on glyphosate use related to each crop type proved to be challenging, as very few data on the use of individual pesticides are recorded and made publicly available in the different countries. The survey results are consistent with references on glyphosate use obtained in the grey literature and in scientific databases (Table 8).

**Table 8.** Comparison of glyphosate use data for different crops and in different countries in some literature references vs. our survey.

| Literature Data | Our Analysis | Consistency Level |
|---|---|---|
| *European and National Level Data* | | |
| *1.* It is estimated that glyphosate is applied on about 30% of arable land [33]. | The survey covered four of the major crops in the European arable land. Within the scope of the survey, 32%, 25%, 52% and 19%, respectively, of the wheat, rape, maize and temporary grassland acreage was treated with glyphosate. | Not possible to check consistency, however there is no contradiction in the data |
| *2.* In Denmark, glyphosate accounted for 35% of all pesticide use in agricultural production in 2009 [34]. | In Denmark, the estimated use of glyphosate was 1241 tonnes a.i. in 2017—i.e., 41% of pesticide sales (3013 t). | Consistent |
| *Data related to annual cropping systems* | | |
| *3.* Glyphosate is the top-ranked herbicide in UK arable crop production [35]. | Glyphosate volumes accounted for 23% of all herbicide a.i. in 2017 in the UK. Among the herbicide a.i. volume in the UK in 2017 (Eurostat), 3% are herbicides H02; 14% are herbicides H03; 3% are herbicides H04; 6% are herbicides H06; 47% are herbicides H99 (including an estimated 23% of glyphosate); 26% are unclassified herbicides. Glyphosate is therefore one of the most used herbicide a.i. in UK. It would be necessary to know the details of the 26% of unclassified herbicides and of the remaining 24% of H99 herbicides in order to conclude if glyphosate is indeed the most commonly used herbicide in the UK. | Consistent |
| *4.* Survey data for the UK revealed a total crop application area of 30% in 2014 [36]. | The percentage of the wheat, rape and maize acreage treated with glyphosate in 2017 was 36%, 71% and 48%, respectively. The percentage of fruit orchards and temporary grassland treated with glyphosate in 2017 was 79% and <1%, respectively. | Not possible to check consistency, however there is no contradiction in the data |

**Table 8.** *Cont.*

| Literature Data | Our Analysis | Consistency Level |
|---|---|---|
| *Data related to annual cropping systems* | | |
| *5.* In Germany, it has been estimated that glyphosate is used on 4.3 million hectares (39%) of agricultural land each year, with nearly two thirds applied to just three crops—oilseed rape, winter wheat and winter barley [37]. | 28% of the maize acreage, 72% of the oilseed rape acreage and 31% of the winter wheat acreage were treated with glyphosate. In terms of acreage, this represents 993,000 ha of winter wheat treated with glyphosate, 708,000 ha of maize treated with glyphosate and 942,000 ha of oilseed rape treated. In total, that is 2.7 million ha treated with glyphosate in just three annual crops. In addition, 53,000 ha of fruit orchards and 70,000 ha of vineyards were treated with glyphosate. In total, 2.8 million ha of five crops were treated with glyphosate. | Consistent |
| *6.* It is estimated that 50% to 60% of sunflower crops in France, Romania and Hungary are treated before harvest with glyphosate [38]. | Sunflower is not included in the survey. As a comparison, in France 35% of the maize acreage, 35% of the oilseed rape acreage and 20% of the winter wheat acreage were treated with glyphosate; in Romania no data was obtained; and in Hungary <1% of the maize acreage, 40–50% of the oilseed rape acreage and 90% of the winter wheat acreage were treated with glyphosate. | Not possible to check consistency |
| *Data related to perennial tree crops* | | |
| *7.* Glyphosate is the most commonly used herbicide in commercial fruit orchards in the UK [39]. | 79% of fruit orchards in UK are treated with glyphosate at least once a year. The average dose is 0.45 kg of a.i./ha. | Not possible to check consistency |
| *8.* Regarding fruit orchards, the number of applications per year varies among crops: on average, within the sample studied, one application per hectare per year for apricot, apple and pear trees; two applications for peach and plum trees; and three for clementines [40]. | According to the survey, the annual dose of glyphosate in fruit orchards varied from 0.45 to 2.50 kg a.i./ha/year. Variations may be due to differences in agronomic practices (e.g., herbicide treatments versus their combination with non-chemical weed control practices), the mode of application (across the whole orchard or only in the row) and different practices depending on the fruit tree species or varieties. | Consistent |

The data on glyphosate use collected through this survey was pieced together from different sources (from national statistics to experts' estimates) and different years (from 2012 to 2017). As a consequence, any comparison of the average use of glyphosate must be considered a preliminary indication and should be refined before further use for other studies or dissemination.

Regarding the data on glyphosate use in annual cropping systems, a specific challenge to data aggregation or comparison arises from the fact that the statistical rules for recording glyphosate treatments applied in the intercropping period vary across countries. This applies to the uses classified as A1, A2 and A4 in Figure 1. Four statistical assignment rules for glyphosate treatments applied in the intercropping period were identified: allocation from harvest to harvest, allocation from field preparation to post-harvest treatments, allocation to the intercropping period and allocation to the cropping system (Table 9). In some countries, several allocation rules may apply depending on the statistical dataset. In this survey, it was not possible to verify the details of data collection in each country. For future monitoring, it is therefore of paramount importance to define common rules for recording data on the use of glyphosate, especially in intercropping periods in annual cropping systems. In annual cropping systems, it might be beneficial to look at the glyphosate use over the entire crop rotation rather than at the individual crop level. The results will be more reliable (as they will not depend on annual allocation rules) and will make it possible to discuss the variability factors. Some factors that explain the level of glyphosate use have already been identified, such as the duration and management of the intercropping period [37,41].

**Table 9.** Statistical rules for the recording of glyphosate treatments applied in the intercropping period in annual cropping systems (before or after an annual crop).

| Allocation Rule | Details |
|---|---|
| **Rule 1.** Allocation from harvest to harvest | After harvest, all treatments are considered as a preparation for the cultivation of the next crop and are therefore allocated to the next crop. Consequently, glyphosate applied after harvest/on stubble is allocated to the next crop. Glyphosate applied right before sowing is allocated to the crop being sown. Desiccation treatments are allocated to the treated crop. |
| **Rule 2.** Allocation from field preparation to post-harvest treatments | Glyphosate applied right before sowing is allocated to the crop being sown. Glyphosate applied after harvest/on stubble is allocated to the crop previously harvested. Desiccation treatments are allocated to the treated crop. |
| **Rule 3.** Allocation to the intercrop period | Glyphosate applied after harvest/on stubble is allocated to the intercrop period. If any application of glyphosate for cover crop destruction (that may also serve for field preparation before the sowing of the next crop), the treatment is also allocated to the intercrop period. |
| **Rule 4.** Allocation to the cropping system | Glyphosate is assumed to benefit all crops in the rotation. It is therefore allocated to the entire crop rotation and not linked to a specific crop. |

Note: in some countries, several allocation rules may apply depending on the statistical dataset.

### 4.3. Factors Explaining the Differences between Countries

Across countries, the AUG varied from 0.04 to 0.56 kg per ha of the UAA, showing considerable variation. Differences were also observed in terms of the acreage of each crop treated with glyphosate annually and the average glyphosate dose applied to a given crop. Factors that may explain the differences in glyphosate uses across countries include: more or less intensive cropping practices; the average farm size (less time for tillage operations/ploughing on larger farms) [37,42]; the percentage of arable cropping area under reduced tillage relying on glyphosate [43]; the percentage of crops with low herbicide inputs (such as permanent grassland) in the UAA; the status of herbicide resistance (e.g., in the UK, the development of herbicide resistance in black-grass resulted in increased glyphosate use [44]); the climatic conditions favourable for weed development; the preference for and availability of other herbicides; the use of non-chemical alternative weed management practices; the use of cover crops; the management of the intercropping period; and the treatment intensity (intra-row versus the whole surface) in perennial crops. Regarding the dose of glyphosate, it must not exceed the maximum application rate stated in the EU and national regulations. However, good agricultural practices and regulations may differ from country to country, which could also partly explain the differences observed. Finally, in annual cropping systems the allocation of glyphosate treatments to a specific crop in the rotation may vary across countries (cf. supra). Further research is needed to untangle the variability in the AUG, the crop acreages treated with glyphosate and the average dose rates of glyphosate across EU countries.

### 4.4. A Framework for Monitoring Glyphosate Use

We propose a framework for the further monitoring of glyphosate use in European agriculture. With several ongoing research projects on glyphosate use in EU countries (e.g., [18,45–48]) and worldwide (e.g., [48]), this framework would help to ensure the consistency and coordination of research activities in EU countries. In addition, this framework will be useful for risk management as well as for designing relevant research and extension activities. The framework encourages a distinction of glyphosate use by the type of crop and agronomic purpose and a clarification of the rationale behind the use (from occasional to recurrent uses) (Figure 1):

- Type of crop, including annual cropping systems, perennial crops and grassland.
- Purpose for which glyphosate is used, including controlling annual or perennial weeds, managing cover crops, ensuring the desiccation of crops or more generally facilitating harvest, terminating temporary grasslands and the renewal of permanent grassland.

- The rationale behind the use of glyphosate includes occasional and recurrent uses. Occasional glyphosate uses are related to exceptional conditions, such as unfavourable meteorological conditions or specific farm constraints. Recurrent glyphosate uses are widespread practices that are embedded in farming systems, while other agronomic solutions may exist but are not employed. Two types of recurrent uses can be distinguished: glyphosate uses related to structural conditions and systematic glyphosate uses not related to structural conditions. Glyphosate uses related to structural conditions apply when the existing farming equipment or infrastructure is not compatible with the use of non-chemical alternative practices (e.g., aboveground irrigation systems in orchards do not allow for mechanical weed control). In contrast, systematic glyphosate uses not related to structural conditions result from the evolution of farming systems characterized by reduced tillage systems; large-scale farms; and the availability of highly efficient, low-priced herbicides such as glyphosate (e.g., the use of glyphosate for winter crop desiccation or the destruction of cover crops).

In total, the framework encompasses eight major agricultural uses of glyphosate: four uses in annual cropping systems (A1 to A4); one type of use in perennial tree crops (P1); and three types of uses in grasslands (G1 to G3).

We also propose a list of 10 indicators that could be recorded and published annually, consistent with the framework. The indicators include general indicators at the national level and indicators of glyphosate use in annual cropping systems, in perennial crops and in grasslands which need to be collected from farms and can then be aggregated at regional or national levels (Table 10).

**Table 10.** List of indicators recommended for a precise monitoring of glyphosate use in European countries.

| Indicator | Unit |
| --- | --- |
| *General INDICATORS* | |
| **I1.** Total annual sales of herbicides | t of herbicide a.i. |
| **I2.** Total annual sales of glyphosate | t of glyphosate a.i. |
| **I3.** Share of herbicide sales to the agricultural sector | t of herbicide a.i. sold to the agricultural sector/total t of herbicide a.i. |
| **I4.** Share of glyphosate versus all herbicides sold to the agricultural sector | t of glyphosate a.i./t of herbicide a.i. |
| **I5.** Average use of glyphosate by the agricultural sector per ha of UAA in conventional agriculture (AUG) | t of glyphosate a.i./ha of UAA in conventional agriculture |
| *Indicators related to annual cropping systems* | |
| **I6.** Percentage of the crop area on which glyphosate is used annually | acreage of the crop in which glyphosate is applied/total acreage of the crop |
| **I7a.** Total use of glyphosate in annual crops | total kg of glyphosate a.i./ha over the crop rotation or for a certain number of years (3–7) |
| **I7b.** Use of glyphosate related to each crop [1] | kg of glyphosate a.i./ha/growing season |
| *Indicators related to perennial tree crops and grassland* | |
| **I8.** Percentage of the crop area on which glyphosate is used annually | acreage of the crop in which glyphosate is applied/total acreage of the crop |
| **I9.** Average use of glyphosate in a given crop | average kg of a.i./ha per treatment |
| **I10.** Total use of glyphosate in a given crop | total kg of a.i./ha per year |

[1] Common rules for the statistical recording of glyphosate treatments occurring in the intercropping period, before or after a crop, should be established at the EU level.

In addition to these indicators, the calculation of a Treatment Frequency Index (TFI) would be a valuable addition. The TFI expresses the number of times a field is treated with the legally approved dose rate of a pesticide. A "herbicide TFI" focuses on herbicide treatments. Summing up all the treatments applied to a crop over the growing season makes it possible to calculate the total TFI at the field level. An average TFI can then be calculated at the farm, regional or national levels. The TFI is

not yet included in the table, as it is not available in many EU countries and different methods may be used for its calculation; however, the proposed indicators provide the necessary information for calculating a TFI with all the relevant details for ensuring comparison between countries. Meanwhile, the dose rate indicators proposed (I7 to I10) already make it possible to show glyphosate application variations, as are visible in Tables 6 and 7 or in other publications [3].

### 4.5. Trends and Factors in the Future Development of Glyphosate Use

To the best of our knowledge, no assessment of the volumes of glyphosate sold in Europe has been conducted within the past 30 years. The data collected in this study over the period 2013 to 2017 are insufficient to assess and discuss the long-term trends in glyphosate use in Europe. Analysts expect future glyphosate sales and markets to grow further, with a forecast annual growth rate for glyphosate sales of about 6% towards 2024, both in Europe [49] and globally [50–52].

Glyphosate is likely to remain a major herbicide, as it has high efficacy, low cost and a wide sprectrum. Some of the factors identified as drivers in the increase in glyphosate use by Benbrook [3] are also relevant in the EU context (e.g., a possible increase in the area treated with glyphosate; the growing adoption of no-tillage and conservation tillage systems relying on glyphosate), whereas other factors do not apply in the current EU context (the commercialization of glyphosate-tolerant crops) or are currently of lower relevance. For example, an increase in the number of authorized glyphosate uses in crops and new application methods are less likely to occur, as glyphosate herbicides are already available for a wide range of crops and uses in Europe. On the other hand, an evolution in the commercial price or taxation of the active ingredient glyphosate (and therefore its affordability and economic competitiveness versus other weed control methods) are key drivers influencing its use in the future. Finally, the evolution of no-tillage and conservation tillage systems might also be a critical aspect, as a large-scale trend towards glyphosate-based conservation agriculture would lead to an increase in glyphosate dependency. In contrast, the development of conservation agriculture systems that rely on alternative management practices for weed control, cover crop management and seedbed preparation could limit the use of glyphosate.

The analysis and monitoring of glyphosate use can only be comprehensive if accompanied by a reflection on the available alternatives. Two types of alternatives to the use of glyphosate exist: other chemical herbicides and non-chemical alternatives, which include preventive and curative methods (mechanical, agronomic, thermic or manual) [53,54]. Their effectiveness, cost and adoption implications for crops and the environment can vary widely or may not be quantified. Further research is needed to assess the conditions, including the economic and technical aspects, as well as the systemic context required for enhancing the adoption of non-chemical alternatives to glyphosate.

Overall, the level of glyphosate use is correlated to the design of agricultural systems in EU countries. The choice of crops and agricultural systems are factors which determine leeway in terms of pesticide use.

**Author Contributions:** Conceptualization, methodology and validation, A.M., C.A. and X.R.; investigation, formal analysis, data curation and writing—original draft preparation, A.C.; writing—review and editing, A.M., C.A., L.U., P.V.B., P.K. and X.R.; supervision, A.M. and P.V.B.; funding acquisition, A.M. All authors have read and agreed to the published version of the manuscript.

**Funding:** This research was funded by INRAE and carried out under the umbrella of the ENDURE network (www.endure-network.eu/what_is_endure/endure_partner_organisations).

**Acknowledgments:** We would like to acknowledge all the professionals who contributed their time to collecting data and providing their expertise to this survey: Andersson R., Auskalnienė O., Barić K., Besenhofer G., Calha I., Carrola Dos Santos S., De Cauwer B., Chachalis D., Dorner Z., Follak S., Forristal D., Gaskov S., Gonzalez Andujar J. L., Hull R., Jalli H., Kierzek R., Kiss J., Lancesseur E., Leonhardt C., Leskovšek R., Lewer A., Mennan H., Nečajeva J., Mullins E., Neve P., Pedraza V., Pintar A., Redl M., Rennick G., Riemens M., Ringselle B., Ruuttunen P., Sattin M., Simić M., Soukup J., Stefanic E., Steinkellner S., Storkey J., Weickmans B. and Wirth J. We thank the three anonymous reviewers for their helpful comments.

**Conflicts of Interest:** The authors declare no conflict of interest.

## Appendix A

**Table A1.** Pesticide sales in the EU28+3 countries in 2017.

| | Total Sales in Tonnes of a.i. | | | | Estimated Use by the Agricultural Sector [1] in kg of a.i./ha [2] | | | |
|---|---|---|---|---|---|---|---|---|
| | Fungicides and Bactericides | Herbicides. Haulm Destructors and Moss Killers | Insecticides and Acaricides | Total | Fungicides and Bactericides | Herbicides. Haulm Destructors and Moss Killers | Insecticides and Acaricides | Total |
| Austria | 1992 | 1297 | 1186 | **4474** | 0.88 | 0.57 | 0.52 | **1.98** |
| Belgium | 2496 | 2334 | 536 | **5366** | 1.80 | 1.69 | 0.39 | **3.88** |
| Bulgaria | 1287 | 1699 | 374 | **3360** | 0.24 | 0.31 | 0.07 | **0.62** |
| Croatia | 727 | 669 | 115 | **1511** | 0.47 | 0.43 | 0.07 | **0.97** |
| Cyprus | 818 | 139 | 124 | **1081** | 6.34 | 1.08 | 0.96 | **8.38** |
| Czech Rep. | 1854 | 2562 | 174 | **4590** | 0.55 | 0.76 | 0.05 | **1.37** |
| Denmark | 484 | 2485 | 44 | **3013** | 0.18 | 0.93 | 0.02 | **1.13** |
| Estonia | 117 | 463 | 26 | **606** | 0.13 | 0.52 | 0.03 | **0.68** |
| Finland | 3228 | 1006 | 25 | **4259** | 1.44 | 0.45 | 0.01 | **1.90** |
| France | 29,770 | 30,230 | 3773 | **63,774** | 0.98 | 0.99 | 0.12 | **2.10** |
| Germany | 13,266 | 16,706 | 14,549 | **44,522** | 0.77 | 0.97 | 0.84 | **2.58** |
| Greece | 1686 | 1674 | 893 | **4252** | 0.32 | 0.32 | 0.17 | **0.81** |
| Hungary | 4171 | 4270 | 860 | **9300** | 0.73 | 0.75 | 0.15 | **1.62** |
| Ireland | 634 | 1820 | 54 | **2508** | 0.13 | 0.37 | 0.01 | **0.51** |
| Italy * | 32,643 | 7114 | 2726 | **42,483** | 2.69 | 0.59 | 0.22 | **3.50** |
| Latvia | 267 | 801 | 33 | **1101** | 0.14 | 0.43 | 0.02 | **0.60** |
| Lithuania | 690 | 1252 | 54 | **1996** | 0.23 | 0.42 | 0.02 | **0.66** |
| Luxembourg * | 91 | 61 | 23 | **175** | 0.65 | 0.44 | 0.16 | **1.25** |
| Malta | 102 | 2 | 3 | **107** | 7.92 | 0.19 | 0.24 | **8.35** |
| Netherlands | 4725 | 2902 | 286 | **7912** | 2.45 | 1.51 | 0.15 | **4.11** |
| Norway * | 93 | 467 | 4 | **564** | 0.09 | 0.45 | 0.00 | **0.54** |
| Poland | 6927 | 13,655 | 1819 | **22,402** | 0.45 | 0.88 | 0.12 | **1.44** |
| Portugal | 4181 | 1899 | 943 | **7024** | 1.12 | 0.51 | 0.25 | **1.89** |
| Romania | 4600 | 5486 | 945 | **11,031** | 0.32 | 0.38 | 0.06 | **0.76** |
| Slovakia | 685 | 1105 | 139 | **1929** | 0.36 | 0.58 | 0.07 | **1.01** |
| Slovenia | 795 | 235 | 50 | **1080** | 1.64 | 0.49 | 0.10 | **2.23** |

**Table A1.** *Cont.*

| | Total Sales in Tonnes of a.i. | | | | Estimated Use by the Agricultural Sector [1] in kg of a.i./ha [2] | | | |
|---|---|---|---|---|---|---|---|---|
| | Fungicides and Bactericides | Herbicides. Haulm Destructors and Moss Killers | Insecticides and Acaricides | Total | Fungicides and Bactericides | Herbicides. Haulm Destructors and Moss Killers | Insecticides and Acaricides | Total |
| Spain * | 37,982 | 16,077 | 6549 | **60,608** | 1.57 | 0.67 | 0.27 | **2.51** |
| Sweden | 265 | 1731 | 31 | **2027** | 0.10 | 0.64 | 0.01 | **0.75** |
| Switzerland | 990 | 599 | 251 | **1840** | 0.65 | 0.39 | 0.16 | **1.21** |
| Turkey | 21,831 | 11,825 | 12,171 | **45,828** | 0.52 | 0.28 | 0.29 | **1.09** |
| UK | 5484 | 9682 | 434 | **15,600** | 0.29 | 0.51 | 0.02 | **0.83** |
| **EU28** | **161,965** | **129,359** | **36,767** | **328,091** | **0.88** | **0.70** | **0.20** | **1.78** |
| **EU28 +3** | **184,880** | **142,251** | **49,192** | **376,324** | **0.81** | **0.62** | **0.21** | **1.64** |

Sources: total sales from Eurostat in 2017 except specific mentions. For Luxembourg and Norway: data from the closest previous year for which data was available. [1] The estimated use by the agricultural sector is calculated with a ratio of 90% (average ratio in the ten countries for which data was available, see Table 1); [2] ha of UAA in conventional agriculture (see Table 1). Scope: EU28+3 includes EU28 countries plus Norway, Switzerland and Turkey.

**Table A2.** Herbicide sales in the EU28+3 countries in 2017 (tons of a.i.).

| | Herbicides. Haulm Destructors and Moss Killers (PES_H) | Herbicides based on Phenoxy-Phytohormones (H01) | Herbicides Based on Triazines and Triazinones (H02) | Herbicides Based on Amides and Anilides (H03) | Herbicides Based on Carbamates and Bis-Carbamates (H04) | Herbicides Based on Dinitroaniline Derivatives (H05) | Herbicides Based on Derivatives of Urea, of Uracil or of Sulphonylurea (H06) | Other Herbicides (Including Glyphosate) (H99) | Unclassified Herbicides |
|---|---|---|---|---|---|---|---|---|---|
| Austria | 1297 | 117 | 187 | 230 | 18 | 54 | 55 | 636 | 0 |
| Belgium | 2334 | 116 | 228 | 302 | 130 | - | 107 | 1393 | 57 |
| Bulgaria | 1699 | | | | | | | | 1699 |
| Croatia | 669 | 18 | 70 | 188 | 9 | 18 | 23 | 343 | 0 |
| Cyprus | 139 | 42 | 1 | 1 | 0 | 1 | 2 | 92 | 0 |
| Czech Rep. | 2562 | 131 | 218 | 598 | 56 | 62 | 208 | 1289 | 0 |
| Denmark | 2485 | 23 | 39 | 99 | 34 | 16 | 9 | 2266 | 0 |
| Estonia | 463 | 61 | 2 | 34 | 0 | 7 | 13 | 333 | 13 |
| Finland | 1006 | 198 | 26 | 7 | | | 7 | 757 | 12 |
| France | 30,230 | 1352 | 709 | 6125 | 391 | 1509 | 1759 | 18,386 | 0 |
| Germany | 16,706 | 1129 | 2374 | 3397 | 237 | 742 | 1175 | 7652 | 0 |

**Table A2.** *Cont.*

| | Herbicides. Haulm Destructors and Moss Killers (PES_H) | Herbicides based on Phenoxy-Phytohormones (H01) | Herbicides Based on Triazines and Triazinones (H02) | Herbicides Based on Amides and Anilides (H03) | Herbicides Based on Carbamates and Bis-Carbamates (H04) | Herbicides Based on Dinitroaniline Derivatives (H05) | Herbicides Based on Derivatives of Urea, of Uracil or of Sulphonylurea (H06) | Other Herbicides (Including Glyphosate) (H99) | Unclassified Herbicides |
|---|---|---|---|---|---|---|---|---|---|
| Greece | 1674 | 80 | 35 | 126 | 2 | 213 | 281 | 937 | 0 |
| Hungary | 4270 | 321 | 307 | 1080 | 8 | 142 | 124 | 2287 | 0 |
| Ireland | 1820 | | | | | | | | 1820 |
| Italy | 7486 | 299 | 370 | 978 | 14 | 332 | 176 | 5319 | 0 |
| Latvia | 801 | 134 | 2 | 109 | 0 | 18 | 16 | 521 | 0 |
| Lithuania | 1252 | 236 | | 191 | | | 32 | 679 | 114 |
| Luxembourg | 61 | 4 | | 12 | 0 | | 2 | 33 | 10 |
| Malta | 2 | | | | | | 1 | 1 | 1 |
| Netherlands | 2902 | | 339 | 332 | 199 | | 51 | 1605 | 376 |
| Norway | 467 | 70 | | | | | | 369 | 29 |
| Poland | 13,655 | 1740 | 728 | | 149 | 250 | | 7989 | 2799 |
| Portugal | 1899 | 55 | 80 | 163 | 2 | 52 | 28 | 1520 | 0 |
| Romania | 5486 | 940 | 281 | 1475 | 23 | 105 | 269 | 2393 | 0 |
| Slovakia | 1105 | 52 | 62 | 232 | 21 | 88 | 60 | 591 | 0 |
| Slovenia | 235 | 9 | 17 | 56 | 0 | 17 | 7 | 129 | 0 |
| Spain | 15,224 | | 277 | 815 | 26 | | 658 | 11,465 | 1984 |
| Sweden | 1731 | | | | | | | | 1731 |
| Switzerland | 599 | 43 | 72 | 87 | 21 | 21 | 36 | 319 | 0 |
| Turkey | 11,825 | 1668 | 162 | 366 | 154 | 830 | 615 | 8031 | 0 |
| UK | 9682 | | 301 | 1390 | 254 | | 610 | 4583 | 2545 |
| **EU28+3** | **141,771** | **8839** | **6885** | **18,392** | **1748** | **4478** | **6324** | **81,915** | **13,189** |
| **Share of the total herbicide sales** | | 6% | 5% | 13% | 1% | 3% | 4% | 58% | 9% |

Data 2017 from Eurostat except specific mentions. Other sources: for Belgium: fytoweb; for Estonia: national statistics (Statistika andmebaas); for Norway: national statistics on pesticide sales. Other years: for Luxembourg, Spain and Italy (data for 2016 instead of 2017) and Norway (average 2014–2018). Scope: EU28+3 includes the EU28 countries plus Norway, Switzerland and Turkey.

## Appendix B

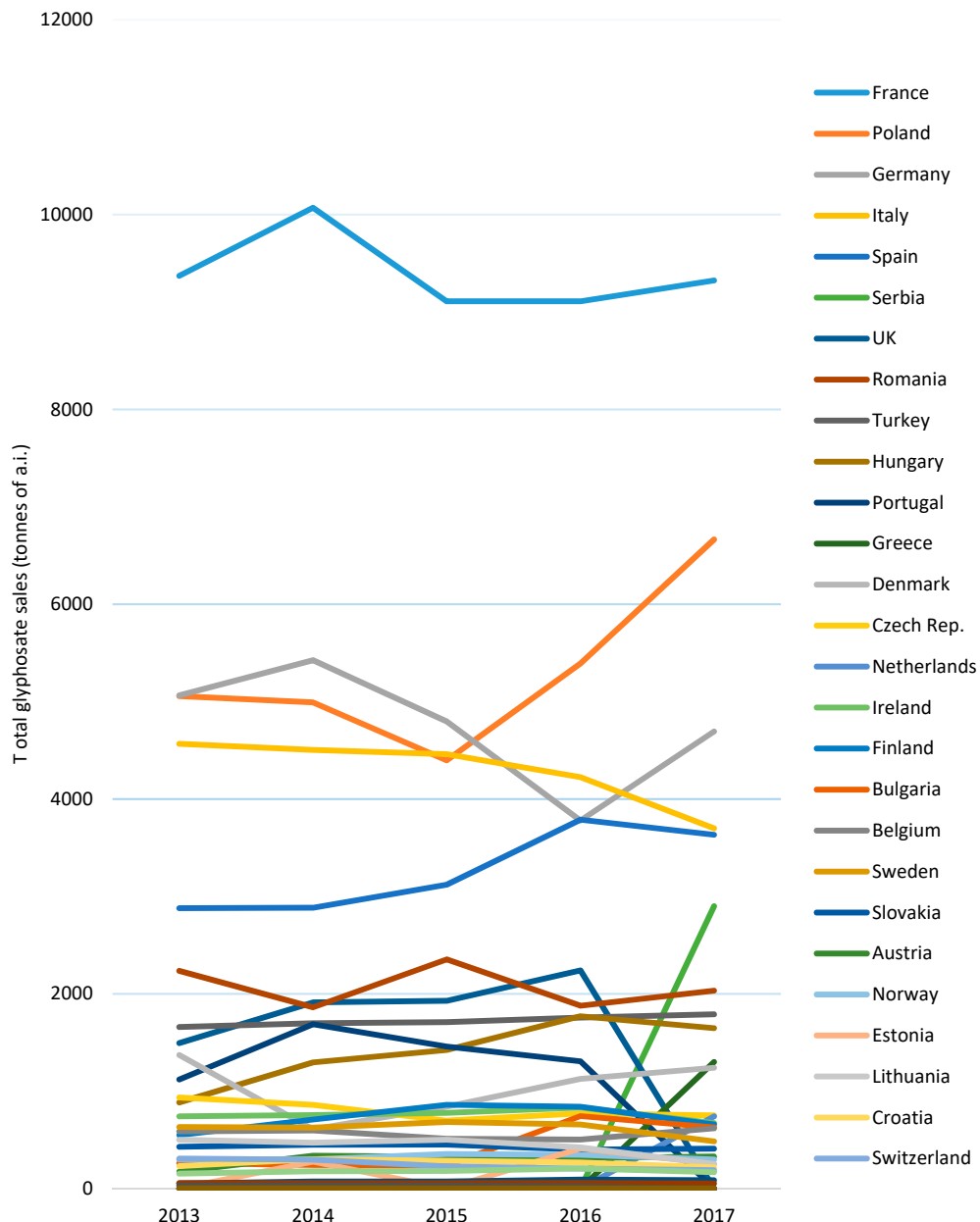

**Figure A1.** Total glyphosate sales in the EU28+4 countries from 2013 to 2017. Survey data and estimated data.

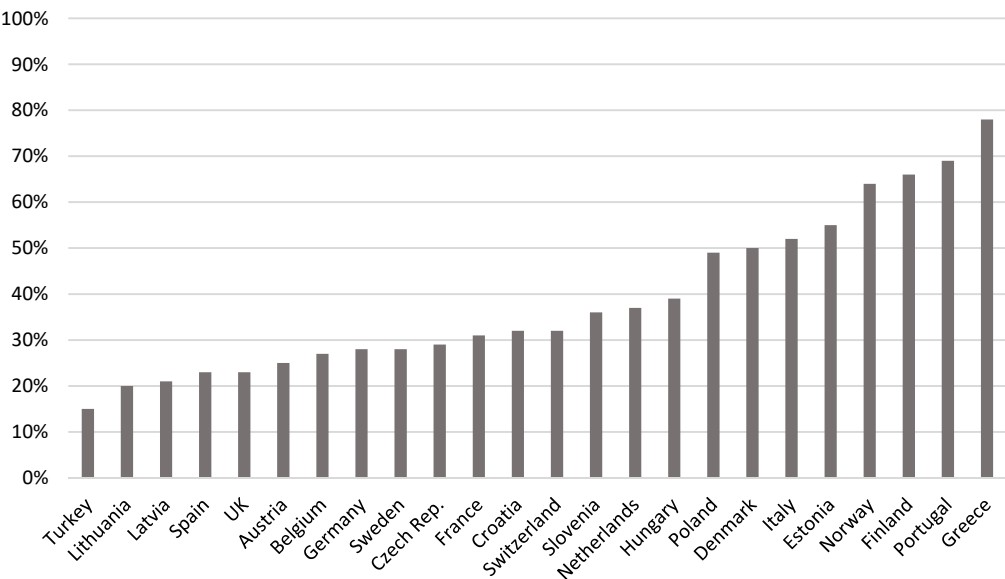

**Figure A2.** Proportion of glyphosate compared to all herbicides in 24 countries in 2017. Only countries for which data was available are represented.

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
