# Peer review of "Glyphosate Use in the European Agricultural Sector and a Framework for Its Further Monitoring"

_sustainability, doi:10.3390/su12145682_

Round 1

Reviewer 1 Report

Excellent article, accurate and extremely relevant, useful both for the academic.

and the political debates.

The article is excellent and could be published as such. But, my advice is to complete it with an analysis that is missing, the development of resistant weeds, its speed, and its link with the observed overuse of glyphosate in some regions.

Even if the companies producing glyphosate promote good practices, they are not always, (not often?) followed by the users.

The authors could make a literature review of what is already available and published on the issue.

It is only a proposal to complete an already very interesting and useful articlawhich deserves to be published withour any doubt?

.

Author Response

Thank you for your review. 

My advice is to complete it with an analysis that is missing, the development of resistant weeds, its speed, and its link with the observed overuse of glyphosate in some regions. Even if the companies producing glyphosate promote good practices, they are not always, (not often?) followed by the users. The authors could make a literature review of what is already available and published on the issue.

The development of resistant weeds indeed is a central topic when discussing the uses of glyphosate. In this article, we focus on the data and could not include a whole literature review on this aspect. However, we added a comment to mention the problem of weed resistance (l. 56) with relevant references.

Reviewer 2 Report

See attached file. 

Author Response

Thank you for your review.

L79 Can you better define these points of contact: how they were identified and selected? How many were in each country? How did they acquire their information?

Additional descriptive details were added (l. 79-81).

L90 Why not be consistent and use one value or the other for comparison?

We provided additional details (l. 92-94).

L99 How consistent are these ratios across countries? Are there variables that may impact this variation (e.g. crops).

This is shown in the results and discussed in the discussion section. In addition, we have added a figure in appendix B to illustrate the results.

L111 Is it important to differentiate these potential exposure pathways? Why the focus on the agriculture sector?

The objective of the research and survey we intended was to study the agricultural uses of glyphosate (in consistency with our research areas). The agricultural sector is the main sector which uses glyphosate (estimated about 90% in volume, both at the EU and global level).

L158 I think it would be interesting to see this data graphically and perhaps temporal changes in glyphosate use over time.

Figure 2 and Figure 3 were added in Appendix B.

L181 I don’t think you need to mention that this will be covered in the discussion.

We deleted the sentence (l. 196).

L197 (Table 3) I think it would be useful for some of this data to also be in graphs so trends can be observed.

Figure 2 and Figure 3 were added in Appendix B.

L264 I don’t think the introductory paragraph is necessary.

We deleted the introductory paragraph (l.284).

Table 8. Just because it is used in most of the fruit orchards does not mean necessarily that it is the most commonly used.

We corrected the conclusion according to your comment.

Reviewer 3 Report

the paper has a lot of data in tables 4-7, but they are dispersed. the research is structured but there are no indicative conclusions on the use of glyphosate.

Author Response

Thank you for your review.

We understand your comment but this paper is the first comprehensive synthesis on glyphosate use in Europe. Data are still missing in some countries and impede the elaboration of a single aggregate value.

Reviewer 4 Report

The paper aimed at presenting new data on glyphosate sales and uses in Europe obtained through a survey carried out within the ENDURE network.

The topic of the paper is important as these data are not known or at least they are not collected all together. Some minor modifications or suggestions are indicated below.

Introduction

As glyphosate is used a lot also in non-crop areas to manage spontaneous vegetation, in the introduction the authors should mention it even though the paper focuses on the uses in agriculture.

In the introduction, a more detailed overview of glyphosate uses in different crops at the Eu level should be added.

Material and Methods

The authors should explain which is the rationale of choosing wheat, maize and rape as representative of annual crops (they were the most treated with glyphosate in UE? the crops most cultivated in the majority of the European countries?…) Please clarify.

Line 160: to help the reader, even though these data are present in the table, please indicate here in which countries glyphosate accounted for more than 50% of the sold herbicides and in which country it accounts for less than 20%.

Results

In Table 6 no all EU countries are present, thus data for maize, rape and wheat were not available for the remaining countries?

paragraph 3.3. the survey analyzed glyphosate uses in wheat, maize and rape. In this paragraph the comments to table 6 should include for which uses (purposes) glyphosate is used in these crops, now they are only indicated in table 6.

Discussion

Table 8. for further literature data to compare with the present survey, the authors could also check the following papers:

Germany: Wiese, A., Schulte, M., Theuvsen, L., Steinmann, H.-H., 2018. Interactions of glyphosate use with farm characteristics and cropping patterns in Central Europe. Pest Manag. Sci. 74, 1155–1165. https://doi.org/10.1002/ps.4542

Italy: Fogliatto, S. Ferrero, A., and Vidotto F., 2020. Current and Future Scenarios of Glyphosate Use in Europe: Are There Alternatives? In Press in Advances in Agronomy, S006521132030050X. https://doi.org/10.1016/bs.agron.2020.05.005. It is a review paper but it contains the estimation that about 30% of arable land are applied with glyphosate.

Figure 1 and other comments on glyphosate uses. The authors could also include in the uses of glyphosate the followings:

  • False seeding technique (to destroy the emerged weeds) (this could be mentioned in the A2 (controlling weeds before cultivation). This practice is quite used in some crops, such as rice in Southern Europe or horticultural crops.
  • A further use of glyphosate is also its application with wiping bars (avoiding crop contact) in rice post-emergence to control weedy rice or barnyardgrass (as these weeds are taller than rice) (see the previous cited paper by Fogliatto et al., 2020). This technique is not only used in rice, but also in horticultural crops or in any crops by using selective equipment (wiper equipment or shielded sprayers to control weed between crop rows), these uses are also indicated in the label of many glyphosate-formulated products.
  •  

Lines 426-434: In the conclusion section, the authors could also indicate that the use of glyphosate will continue to increase as glyphosate has high efficacy, low cost, and wide-sprectrum. Also the non-chemical alternatives can be applied but in some cases they required repeated applications (mechanical weeding) or combination of techniques that can be more costly (see different papers in the literature).

Author Response

Thank you for your comprehensive and positive review.

  1. In the Introduction

As glyphosate is used a lot also in non-crop areas to manage spontaneous vegetation, in the introduction the authors should mention it even though the paper focuses on the uses in agriculture.

A comment was added in this regard (l. 35).

In the introduction, a more detailed overview of glyphosate uses in different crops at the EU level should be added.

A sentence was added to provide this overview (l. 46).

The authors should explain which is the rationale of choosing wheat, maize and rape as representative of annual crops (they were the most treated with glyphosate in UE? the crops most cultivated in the majority of the European countries? …) Please clarify.

in order to cover a diversity of situations and a significant share of the European acreage, one cereal (wheat) and two other major European annual crops (maize and rape), perennial tree crops (olive groves, vineyards and fruit orchards) as well as temporary grasslands were included in the survey in order to cover a diversity of situations and a significant share of the European acreage. Together, the six crops and temporary grassland covered 36% of the total EU28 UAA in 2017 (Eurostat). This was added in l. 143-146.

Line 160: to help the reader, even though these data are present in the table, please indicate here in which countries glyphosate accounted for more than 50% of the sold herbicides and in which country it accounts for less than 20%.

This indication was added in l. 170-173.

In Table 6 no all EU countries are present, thus data for maize, rape and wheat were not available for the remaining countries?

Indeed, data for maize, rape and wheat were not available for the remaining countries. The title of the table (l. 250) was adjusted and a comment was added in the table footnote (l. 266-267).

Table 7.

The formatting of the table was adjusted to make clearer than only the last column refers to temporary grasslands.

Table 8. for further literature data to compare with the present survey, the authors could also check the following papers: Wiese, A., Schulte, M., Theuvsen, L., Steinmann, H.-H., 2018. Interactions of glyphosate use with farm characteristics and cropping patterns in Central Europe. Pest Manag. Sci. 74, 1155–1165. https://doi.org/10.1002/ps.4542; Italy: Fogliatto, S. Ferrero, A., and Vidotto F., 2020. Current and Future Scenarios of Glyphosate Use in Europe: Are There Alternatives? In Press in Advances in Agronomy, S006521132030050X. https://doi.org/10.1016/bs.agron.2020.05.005. It is a review paper but it contains the estimation that about 30% of arable land are applied with glyphosate.

These are indeed relevant articles. The reference to Fogliatto et al was added in the table 8 (see [32]) and discussed. The reference to Wiese et al was added in the text (l.470, see [42]). Thanks to this article, we identify an additional reference (ss [35] that we included in the analysis in Table 8.

Figure 1 and other comments on glyphosate uses. The authors could also include in the uses of glyphosate the followings:

  • False seeding technique (to destroy the emerged weeds) (this could be mentioned in the A2 (controlling weeds before cultivation). This practice is quite used in some crops, such as rice in Southern Europe or horticultural crops.

That was added in the Figure 1 footnote (l. 435-436).

  • A further use of glyphosate is also its application with wiping bars (avoiding crop contact) in rice post-emergence to control weedy rice or barnyardgrass (as these weeds are taller than rice) (see the previous cited paper by Fogliatto et al., 2020). This technique is not only used in rice, but also in horticultural crops or in any crops by using selective equipment (wiper equipment or shielded sprayers to control weed between crop rows), these uses are also indicated in the label of many glyphosate-formulated products.

That was added in the Figure 1 footnote (l. 436-438).

Lines 426-434: In the conclusion section, the authors could also indicate that the use of glyphosate will continue to increase as glyphosate has high efficacy, low cost, and wide-sprectrum.

The comment was added (l.450-451).

Also the non-chemical alternatives can be applied but in some cases they required repeated applications (mechanical weeding) or combination of techniques that can be more costly (see different papers in the literature).

A sentence was added (l.468-469).

Round 2

Reviewer 3 Report

This paper provides a database on glyphosate use in EU countries, for the entire agricultural sector.  The data, reported in tables, cover the period 2013-2017, didn't permitt to discuss long-term trends in glyphosate usa in Europe. Further research is needed to assess the economic and technical aspects, as well as the ecologic context relative to the alternative use of glyphosate.  The choice of crops and agricultural systems are factors which determine the pesticide use.

Tables are numerous and sometimes dispersed. I understand the need of the authors because the data is many and you can not combine by presenting fewer tables. 

I appreciate the effort to present data also in the form of histograms or pictures, for easy visualization.